# Utilization of health insurance by patients with diabetes or hypertension in urban hospitals in Mbarara, Uganda

Peter Kangwagye[1,2]*, Laban Waswa Bright[1], Gershom Atukunda[1], Robert Basaza[3], Francis Bajunirwe[4]

1 Department of Public Health, Bishop Stuart University, Mbarara, Uganda, 2 Department of Internal Medicine, Mbarara University of Science and Technology, Mbarara, Uganda, 3 Department of Public Health, Leadership Program, Uganda Christian University, Kampala, Uganda, 4 Department of Community Health, Mbarara University of Science and Technology, Mbarara, Uganda

* peterkangwagye@gmail.com

## Abstract

### Background

Diabetes and hypertension are among the leading contributors to global mortality and require life-long medical care. However, many patients cannot access quality healthcare due to high out-of-pocket expenditures, thus health insurance would help provide relief. This paper examines factors associated with utilization of health insurance by patients with diabetes or hypertension at two urban hospitals in Mbarara, southwestern Uganda.

### Methods

We used a cross-sectional survey design to collect data from patients with diabetes or hypertension attending two hospitals located in Mbarara. Logistic regression models were used to examine associations between demographic factors, socio-economic factors and awareness of scheme existence and health insurance utilization.

### Results

We enrolled 370 participants, 235 (63.5%) females and 135 (36.5%) males, with diabetes or hypertension. Patients who were not members of a microfinance scheme were 76% less likely to enrol in a health insurance scheme (OR = 0.34, 95% CI: 0.15–0.78, p = 0.011). Patients diagnosed with diabetes/hypertension 5–9 years ago were more likely to enrol in a health insurance scheme (OR = 2.99, 95% CI: 1.14–7.87, p = 0.026) compared to those diagnosed 0–4 years ago. Patients who were not aware of the existing schemes in their areas were 99% less likely to take up health insurance (OR = 0.01, 95% CI: 0.0–0.02, p < 0.001) compared to those who knew about health insurance schemes operating in the study area. Majority of respondents expressed willingness to join the proposed national health insurance scheme although concerns were raised about high premiums and misuse of funds which may negatively impact decisions to enrol.

**Data Availability Statement:** All the necessary data is provided within the paper.

**Funding:** The authors received no specific funding for this work.

**Competing interests:** The authors have declared that no competing interests exist.

## Conclusion

Belonging to a microfinance scheme positively influences enrolment by patients with diabetes or hypertension in a health insurance program. Although a small proportion is currently enrolled in health insurance, the vast majority expressed willingness to enrol in the proposed national health insurance scheme. Microfinance schemes could be used as an entry point for health insurance programs for patients in these settings.

## Introduction

Chronic diseases like hypertension and diabetes rank among the leading causes of illness for all ages in Uganda [1]. In urban Mbarara and rural southwest Uganda, many patients often fail to keep up with the treatment regime because they are unable to meet the high costs of health care [2]. Sadly, the national health insurance scheme (NHIS) which would have offered protection against the risk of incurring high costs of health services has not yet been established in the country. More critically, there has been limited participation of potential contributors and beneficiaries of the scheme, including patients with diabetes or hypertension, in the search for best approaches to ensure that the goals of the proposed NHIS are achieved and sustained. This study thus sought to provide information on the views and inputs of a critical group of intended beneficiaries when the country is still debating how to design its universal health coverage (UHC) roadmap [3].

Treatable and chronic diseases reduce a household's income if people are not able to work. If they fail to raise the funds to pay for medical fees, people in rural and urban areas may end up foregoing treatment even when they know that this behaviour may negatively impact their long-term health. Also, households may be forced to sell their productive assets and take expensive loans so as to be able to pay for medical care. In a study of coping strategies in Uganda, Leliveld [4] reported how households sold land, cattle, or goats or used their savings to respond to long-term illness. Other studies in Uganda, Kenya and Ethiopia have gone further and demonstrated how the costs of illness contribute significantly to the impoverishment of households in rural and urban areas [5–7]. Such huge medical expenses which ultimately impoverish people are referred to as catastrophic out-of-pocket expenditures (OOP) and are known to constitute a critical impediment to achieving universal health coverage [8–10].

Health insurance is a form of insurance that covers the whole or a part of the risk of a person incurring medical expenses, spreading the risk over numerous people [11–13]. It is a critical pillar in resource mobilization for achieving universal health coverage since individuals or households that pay a certain predetermined amount of money in return receive a health-care benefit package covering them and their dependants. This prepayment mechanism is particularly useful for cushioning households from catastrophic poverty, injury and death resulting from treatable and chronic diseases [14–17]. Studies show that microfinance schemes could potentially act as intermediaries in this effort as they could put in place financing mechanisms which favour a reduction in healthcare costs for their members [18, 19].

To date, no study has been undertaken to ascertain the proportion of patients with diabetes or hypertension in urban Mbarara who are currently enrolled in social, private and community-based health insurance schemes and those who are willing to enrol in the proposed NHIS. Further, an in-depth inquiry into how demographic factors, socio-economic factors and awareness of scheme existence influence utilization of health insurance has not been undertaken. Therefore, this study aimed to fill the identified knowledge gaps by assessing the factors

affecting the utilization of health insurance by patients receiving diabetes and hypertension care in the study area.

## Methods

### Ethics statement

Approval to conduct the study was obtained from the directorate of graduate studies, research, and innovations at Bishop Stuart University. Further approval was obtained from the hospital directors of Mbarara Regional Referral Hospital and Divine Mercy Hospital. Written informed consent was obtained from study participants aged 18 years and older and assurance given that the information obtained will be treated with confidence and that at all times we will present the data in such a way that the participant's identity cannot be connected with any specific responses.

### Study setting

The study was based at Mbarara Regional Referral Hospital (MRRH) and Divine Mercy Hospital (DMH), two hospitals, both located in urban Mbarara. MRRH is public and DMH is a private facility and both serve as referral sites for a large number of patients with diabetes or hypertension in south-western Uganda, a region of at least 5 million people. Health facility records show that about 5,000 patients seek diabetes and hypertension care at these hospitals each month. When compared to other regions, western Uganda has the second highest prevalence of hypertension in the country estimated at 32.5% [20]. The prevalence of diabetes has been estimated at 8.1% and 9.0% for Kampala in the central region and Kasese in mid-western Uganda, respectively [21, 22].

### Participant recruitment

Study participants were selected using systematic sampling. First, we estimated the population seeking diabetes and hypertension care in each hospital per month, and then dividing this number by the required sample size to obtain the sampling interval. Preliminary review of the available medical records led to a sampling interval of every 3rd patient from each diabetes/ hypertension clinic being selected to participate in the study, with the first patient who came for care each day being the starting point.

We used a sample size estimation formula by Slovin (1960) to determine the required sample size since the estimated population of patients with diabetes or hypertension was known. We assumed the population size to be 5,000, and 0.05 as the level of precision. The calculation yielded a sample size of 370 participants.

### Data collection

Data were collected from study participants for a period of three months, namely, between May and July 2020. At MRRH the hypertension and diabetes clinics are operated once a week on Tuesdays and Thursdays, respectively, while at DMH the clinics have no specific days. We deployed two research assistants at each clinic to identify eligible participants and administer questionnaires to the respondents.

**Study tool.** Each questionnaire comprised both closed and open ended questions regarding demographic factors, socio-economic factors and awareness of scheme existence. It also comprised questions about the proposed NHIS. Specifically, the study participants were asked whether they believed it was important to have a NHIS in Uganda, if they would be willing to join the proposed scheme. Participants were also asked about what they thought should be

options for those who would not be able to join the scheme due to their inability to pay. The participants were also asked if they were willing to contribute to the proposed scheme, how much they would be willing to contribute and how frequently they would want to make contributions. Participants were interviewed separately and their names excluded to guarantee privacy during interviewing and confidentiality of information obtained.

**Quality control.** Prior to data collection, we pre-tested the questionnaire for consistency and suitability at a non-participating hospital in Mbarara town. Thirty seven pilot respondents, representing 10% of sample size of 370, were interviewed. The comments and suggestions from the pilot study were used to revise the tool and ensure questions were understandable. During data collection, debriefing meetings were held at the end of each day to review data and identify any omissions and errors.

## Data management and analysis

The completed questionnaires were examined by the first author to confirm completeness and consistency. The data were then entered and cleaned using Microsoft Excel and backed up on an external hard drive. Quantitative data were analysed using Microsoft Excel and Minitab software package version 14.

The primary outcome for this study was utilization of health insurance. We defined this as the proportion of diabetes/hypertension patients enrolled in any health insurance scheme at the time of conducting this study. The secondary outcome was willingness to participate in the NHIS, which refers to whether or not people were willing to join the proposed NHIS. Univariate, bivariate and multivariable analyses were used to examine the association between utilization of health insurance and the demographic and socio-economic factors and awareness of scheme existence. Variables with a p-value less than 0.05 in bivariate analyses were selected for inclusion in multivariate logistic regression models. The odds ratios associated with these factors were then reported as a measure of strength, together with the respective 95% confidence intervals and p-values.

## Results

### Demographic and socio-economic factors

A total of 370 diabetes and hypertension patients attending Mbarara Regional Referral Hospital and Divine Mercy Hospital participated in the study after approaching 387 patients; yielding a response rate of 96%. Out of 370 participants, 135 (36.5%) were male and 235 (63.5%) were female. Participants who had attained primary education were the majority (41.6%) followed by those who had never had formal education (24.9%). Their mean age was 57.3 years, with the youngest aged 18 and the oldest 89 years. Household size ranged between one and 20 members and consisted of an average of 6.3 members (see Table 1).

### Awareness of and enrolment in health insurance schemes

As shown in Table 2, the majority of respondents (50.6%) had never heard about health insurance schemes operating in the study area. A larger majority (59.2%) had never been enrolled in any health insurance scheme and even had not currently been enrolled in any health insurance scheme (58.0%) at the time of conducting this study. Most of those who currently had health insurance cover were members of patient-driven associations (88.7%), Jubilee (2.6%) and UAP Old Mutual Uganda (2.0%) insurance schemes. Asked how frequently they were required to pay premiums, the majority (96.1%) indicated that annual contribution was the most commonly used frequency of making contributions to insurance schemes. They also

**Table 1. Demographic and socio-economic characteristics of study participants.**

| Variable | Category | n (%) |
|---|---|---|
| Gender | Male | 135 (36.5%) |
| | Female | 235 (63.5%) |
| Age (years) | Mean; range | 57.3; 18–89 |
| Marital status | Married | 297 (80.3%) |
| | Widow | 48 (13.0%) |
| | Divorced | 12 (3.2%) |
| | Single | 13 (3.5%) |
| Household size | Mean; range | 6.3; 1–30 |
| Level of education | None | 92 (24.9%) |
| | Primary | 154 (41.6%) |
| | Secondary | 67 (18.1%) |
| | Tertiary | 57 (15.4%) |
| Main source of income | Farming | 232 (64.8) |
| | Business enterprises | 72 (20.1) |
| | Salaried employment | 54 (15.1) |
| Income per month | Below Ug Shillings 100,000 | 207 (57.0%) |
| | Ug Shillings 100,000–500,000 | 120 (33.1%) |
| | Ug Shillings 500,000–1,000,000 | 26 (7.2%) |
| | Above Ug Shillings 1,000,000 | 10 (2.8%) |
| Member of a microfinance scheme | Yes | 148 (41.3%) |
| | No | 210 (58.7%) |
| Years since respondents were first diagnosed with diabetes/hypertension | Mean; range | 6.6; 0–41 |
| Perception about diabetes/hypertension | Extremely dangerous | 334 (90.3%) |
| | Somewhat dangerous | 17 (4.6%) |
| | Not at all dangerous | 15 (4.1%) |
| | I don't know | 4 (1.1%) |
| Able to keep paying OOP | Yes | 209 (62.8%) |
| | No | 124 (37.2%) |

OOP = Out-Of-pocket Payments.

cited the reduced cost of drugs (88.7%) as the main reason for their choice of a particular health insurance scheme. In terms of persons covered by insurance, most of the respondents (92.2%) said that only the insured person was covered. The majority of those who had not been enrolled in any health insurance scheme (63.9%) cited lack of information as the main impediment. When asked how they were coping, the non-enrolled (45.3%) cited help from family and friends as the main fall-back alternative.

**Table 2. Descriptive statistics of the participants' awareness of health insurance schemes.**

| Variable | Category | n (%) |
|---|---|---|
| Heard about health insurance schemes | Yes | 178 (49.4%) |
| | No | 182 (50.6%) |
| Had ever been enrolled in a health insurance scheme | Yes | 155 (42.0%) |
| | No | 214 (58.0%) |
| Currently enrolled in any health insurance scheme | Yes | 149 (40.8%) |
| | No | 216 (59.2%) |
| Frequency of paying premiums | Annually (every 12 months) | 146 (96.7%) |
| | Quarterly (every 3 months) | 2 (1.3%) |
| | Monthly (every month) | 3 (2.0%) |
| Main reason for choosing a particular scheme | Reduced cost of drugs | 134 (88.2%) |
| | Regular access to medicines | 4 (2.6%) |
| | To avoid having to pay each time they visit | 1 (0.7%) |
| | Helps in times of emergency | 1 (0.7%) |
| | Has aspect of education for children | 1 (0.7%) |
| | Choice made by others | 11 (7.2%) |
| Persons covered by insurance | Insured person alone | 130 (92.9%) |
| | Insured person and spouse | 4 (2.9%) |
| | Insured person, spouse and up to four children | 6 (4.3%) |
| | Insured person, spouse and all dependants | 0 (0%) |
| Main reason for non-enrolment | Lack of information on health insurance | 53 (64.6%) |
| | High insurance premiums | 22 (26.8%) |
| | Mistrust of health insurance agents | 2 (2.4%) |
| | No need for health insurance | 5 (6.1%) |
| Coping strategies | Salary from last month | 8 (3.7%) |
| | Savings | 48 (22.4%) |
| | Help from family and friends | 97 (45.3%) |
| | Loans | 1 (0.5%) |
| | Reduction in daily living cost | 60 (28.0%) |

## Factors associated with utilization of health insurance

Logistic regression at a bivariate level revealed that utilization of health insurance was significantly associated with household size, income source, monthly income, membership of a microfinance scheme, years since the first time when the patient was first diagnosed with diabetes or hypertension and awareness about the existence of health insurance schemes (Table 3). Individuals who had families of 5–8 members were 1.7 times more likely to enrol in a health insurance scheme compared to those who had families of 1–4 members (OR = 1.67, 95% CI: 1.03–2.7, p = 0.038) while those that had families of > 8 members were 2.3 times likely to enrol in a health insurance scheme compared to those who had families of 1–4 members (OR = 2.28, 95% CI: 1.28–4.08, p = 0.005). Individuals who depended on salaried employment were 2.4 times more likely to enrol in a health insurance scheme compared to those who depended on farming (OR = 2.43, 95% CI: 1.31–4.52, p = 0.005).

The participants whose income was between 100,000 and 500,000 shillings per month were 1.6 times more likely to enrol in a health insurance scheme compared to those who earned less than 100,000 shillings per month (OR = 1.59, 95% CI: 1.0–2.53, p = 0.049) while those who earned an income between 500,000 and 1,000, 000 shillings per month were 3.4 times more likely to enrol in a health insurance scheme compared to who earned less than 100,000

**Table 3. Logistic regression of factors associated with utilization of health insurance (n = 370).**

| | Bivariate analysis | | Multivariable analysis | |
|---|---|---|---|---|
| Independent variable | OR (95% CI) | P-value | OR (95% CI) | P-value |
| Age (years) | | | | |
| 18–44 | 1 | | | |
| 45–54 | 0.73 (0.37–1.42) | 0.350 | | |
| 55–64 | 1.11 (0.56–2.17) | 0.771 | | |
| 65–89 | 0.75 (0.40–1.43) | 0.381 | | |
| Gender | | | | |
| Male | 1 | | | |
| Female | 0.76 (0.49–1.17) | 0.207 | | |
| Marital status | | | | |
| Married | 1 | | | |
| Single/Divorced/Widowed | 0.75 (0.43–1.28) | 0.285 | | |
| Household size | | | | |
| 1–4 members | 1 | | 1 | |
| 5–8 members | 1.67 (1.03–2.7) | **0.038** | 1.87 (0.76–4.57) | 0.172 |
| > 8 members | 2.28 (1.28–4.08) | **0.005** | 1.64 (0.60–4.50) | 0.340 |
| Level of education | | | | |
| None | 1 | | | |
| Primary | 0.67 (0.39–1.14) | 0.137 | | |
| Secondary | 1.42 (0.75–2.68) | 0.284 | | |
| Tertiary | 1.11 (0.57–2.18) | 0.759 | | |
| Main source of income | | | | |
| Farming | 1 | | 1 | |
| Business enterprises | 1.42 (0.82–2.46) | 0.211 | 0.97 (0.32–2.91) | 0.955 |
| Salaried employment | 2.43 (1.31–4.52) | **0.005** | 1.61 (0.48–5.36) | 0.440 |
| Income per month | | | | |
| Below Shillings 100,000 | 1 | | 1 | |
| 100,000–500,000 | 1.59 (1.0–2.53) | **0.049** | 0.69 (0.26–1.80) | 0.450 |
| 500,000–1,000, 000 | 3.4 (1.43–8.09) | **0.006** | 0.42 (0.09–1.95) | 0.270 |
| Above 1,000,000 | 1.91 (0.54–6.84) | 0.317 | 0.21 (0.03–1.68) | 0.140 |
| Member of a microfinance scheme | | | | |
| Yes | 1 | | 1 | |
| No | 0.34 (0.22–0.53) | **< 0.001** | 0.34 (0.15–0.78) | **0.011** |
| Years since first diagnosis | | | | |
| 0–4 | 1 | | 1 | |
| 5–9 | 2.87 (1.69–4.9) | **< 0.001** | 2.99 (1.14–7.87) | **0.026** |
| 10–41 | 3.15 (1.85–5.36) | **< 0.001** | 2.21 (0.86–5.69) | 0.102 |
| Perception about diabetes/hypertension | | | | |
| Extremely dangerous | 1 | | | |
| Somewhat dangerous | 0.98 (0.36–2.64) | 0.970 | | |
| Not at all dangerous | 0.51 (0.16–1.63) | 0.257 | | |
| I don't know | 0.47 (0.05–4.54) | 0.512 | | |
| Able to keep paying OOP | | | | |
| Yes | 1 | | | |
| No | 0.7 (0.44–1.11) | 0.131 | | |
| Aware of existing health insurance schemes | | | | |
| Yes | 1 | | 1 | |
| No | 0.01 (0.0–0.07) | **< 0.001** | 0.01 (0.0–0.02) | **< 0.001** |

shillings per month (OR = 3.4, 95% CI: 1.43–8.09, p = 0.006). Participants who had not been members of a microfinance scheme were 78% less likely to enrol in a health insurance scheme (OR = 0.32, 95% CI: 0.14–0.75, p = 0.011).

Participants who had been diagnosed with diabetes/hypertension 5–9 years ago were 2.9 times more likely to enrol in a health insurance scheme (OR = 2.87, 95% CI: 1.69–4.9, p < 0.001) than those who had been diagnosed with the chronic condition 0–4 years ago while those that had been diagnosed with diabetes/hypertension 10–41 years ago were 3.2 times more likely to enrol in a health insurance scheme (OR = 3.15, 95% CI: 1.85–5.36, p < 0.001) compared to those who had been diagnosed with the chronic condition 0–4 years ago. Lastly, the bivariate analysis revealed that patients who were not aware of the existing schemes were 99% less likely to take up health insurance (OR = 0.01, 95% C I: 0.0–0.07, p < 0.001) compared to those who knew about existing insurance schemes.

The multivariable analysis showed that participants who had not been members of a microfinance scheme were 76% less likely to enrol in a health insurance scheme (OR = 0.34, 95% CI: 0.15–0.78, p = 0.011, see Table 3). Participants who had been diagnosed with diabetes/hypertension between 5–9 years ago were almost three times more likely to enrol in a health insurance scheme (OR = 2.99, 95% CI: 1.14–7.87, p = 0.026) than those who had been diagnosed with the chronic condition 0–4 years ago. The multivariable analysis also showed that patients who were not aware of the existing schemes were 99% less likely to take up health insurance (OR = 0.01, 95% CI: 0.0–0.02, p < 0.001) compared to those who knew about health insurance schemes operating in the district.

## Willingness to enrol in the proposed NHIS

When asked if they had heard about the proposed NHIS, the majority (80.9%) said that they had not heard about it (Table 3). And when asked whether NHIS would be good for Uganda, the majority (97.5%) answered "yes". A large fraction (i.e. 42.9% of the respondents) said that the main reason why they considered NHIS to be a good idea is that the scheme will help people to save money incurred on paying for treatment which is often expensive. Only three people had reservations about the scheme and cited high premiums, corruption and the absence of need to join another insurance scheme as the main reasons for this view (Table 4).

The respondents who said they would like to join the proposed scheme were the majority at 97.8%, and went ahead to propose that premiums should be made every month and that each person should, on average, contribute 8,200 shillings per month. Regarding those who will not be able to join because they are indigent, the respondents suggested that these should be exempt from payment and that their contributions should be paid by government or donors. They also suggested that if not exempt, indigent persons should pay subsidized premiums and be encouraged to join savings and credit schemes so as to be able to afford subsidized premiums. They further indicated that improvement of services at government facilities would cater for concerns about indigent persons as this would ensure quality care for all.

At a bivariate level, logistic regression showed that willingness to enrol in the proposed NHIS was significantly associated with income source and the income earned per month (Table 5). Individuals who depended on salaried employment were 85% less likely to be willing to enrol in the proposed NHIS compared to those who depended on farming (OR = 0.15, 95% CI: 0.02–0.93, p = 0.042). The participants who earned more than 500,000 shillings (about USD 142.9) per month were 94% less likely to be willing to enrol in the proposed NHIS as compared to those whose income was less than 100,000 shillings (about USD 28.6) per month (OR = 0.09, 95% CI: 0.01–0.97, p = 0.047). However none of these two variables remained statistically significant when a multivariable analysis was performed.

**Table 4. Descriptive statistics of the participants' willingness to enrol in the proposed National Health Insurance Scheme (NHIS).**

| Variable | Category | n (%) |
|---|---|---|
| Had heard about the proposed NHIS | Yes | 67 (19.1%) |
| | No | 284 (80.9%) |
| Is it a good idea to have the NHIS in Uganda? | Yes | 345 (97.5%) |
| | No | 9 (2.5%) |
| Why it is good or not good to have NHIS in Uganda | Will help people to save money incurred on paying for treatment which is often expensive | 158 (42.9%) |
| | Will ease access to quality health care | 87 (23.6%) |
| | Will lead to improved health seeking behaviour as a result of early screening | 52 (14.1%) |
| | Will bring about equity because it will act as a risk-pooling mechanism | 68 (18.5%) |
| | NHIS will have high premiums | 1 (0.3%) |
| | Funds remitted to government will not be used properly due to corruption | 1 (0.3%) |
| | No need to join another insurance scheme | 1 (0.3%) |
| Willing to join the proposed NHIS | Yes | 316 (97.8%) |
| | No | 7 (2.2%) |
| Proposed contribution per month (in shillings) | Mean; range | 8,200; 500–100,000 |
| Frequency of making contributions | Annually (every 12 months) | 133 (36.7%) |
| | Semi-annually (every 6 months) | 9 (2.5%) |
| | Quaterly (every 3 months) | 21 (5.8%) |
| | Monthly (every month) | 190 (52.5%) |
| | Weekly (every week) | 9 (2.5%) |
| Suggestions for those who will not be able to join because they are unable to pay (poorest members) | They should be exempt from payment | 259 (76.2%) |
| | Their contributions should be paid by government or donors | 37 (10.9%) |
| | Services at government facilities should be improved to ensure quality care for all | 28 (8.2%) |
| | They should pay subsidized premiums | 6 (1.8%) |
| | They should be encouraged to join savings and credit schemes to afford subsidized premiums | 3 (0.9%) |
| | I don't know | 7 (2.1%) |

## Discussion

### Socio-economic factors

This study found that utilization of health insurance by patients with diabetes or hypertension was low, standing at only 40.8%. The utilization was associated with participants' membership of a microfinance scheme and years since the first time when they were first diagnosed with diabetes/hypertension. Patients who had not been members of a microfinance scheme were less likely to enrol in a health insurance scheme. A potential explanation for this finding may be the potential role played by microfinance schemes in improving social solidarity and supporting their members to get used to making regular small contributions. Microfinance schemes may thus be used for priming participants to the behaviour of regular payments when designing and implementing the forthcoming NHIS. Studies in the Phillipines have observed that microfinance schemes when combined with health can enhance service delivery and uptake of policies [18]. In Kenya microfinance schemes have put in place health care financing

**Table 5. Logistic regression of factors associated with the willingness of participants to enrol in the proposed national health insurance scheme.**

| | Bivariate analysis | | Multivariable analysis | |
|---|---|---|---|---|
| Independent variable | OR (95% CI) | P-value | OR (95% CI) | P-value |
| Age (years) | | | | |
| 18–44 | 1 | | | |
| 45–54 | 0.95 (0.08–10.72) | 0.965 | | |
| 55–64 | 1.57 (0.10–25.78) | 0.750 | | |
| 65–89 | 0.75 (0.08–7.42) | 0.807 | | |
| Gender | | | | |
| Male | 1 | | | |
| Female | 2.21 (0.49–10.03) | 0.305 | | |
| Marital status | | | | |
| Married | 1 | | | |
| Single/Divorced/Widowed | 0.55 (0.10–2.91) | 0.482 | | |
| Household size | | | | |
| 1–4 members | 1 | | | |
| 5–8 members | 1.72 (0.28–10.47) | 0.556 | | |
| > 8 members | 0.85 (0.14–5.24) | 0.864 | | |
| Level of education | | | | |
| None | 1 | | | |
| Primary | 0.85 (0.08–9.50) | 0.894 | | |
| Post-primary | 0.33 (0.04–2.97) | 0.320 | | |
| Main source of income | | | | |
| Farming | 1 | | | |
| Business enterprises | 0.25 (0.03–1.83) | 0.172 | 0.43 (0.05–3.54) | 0.433 |
| Salaried employment | 0.15 (0.02–0.93) | **0.042** | 0.37 (0.05–2.84) | 0.336 |
| Income per month | | | | |
| Below Shillings 100,000 | 1 | | | |
| 100,000–500,000 | 0.14 (0.01–1.23) | 0.076 | 0.22 (0.02–2.48) | 0.222 |
| Above 500,000 | 0.09 (0.01–0.97) | **0.047** | 0.17 (0.01–2.58) | 0.200 |
| Member of a microfinance scheme | | | | |
| Yes | 1 | | | |
| No | 3.82 (0.73–20.02) | 0.112 | | |
| Years since first diagnosis | | | | |
| 0–4 | 1 | | | |
| 5–9 | 1.16 (0.12–11.3) | 0.901 | | |
| 10–41 | 0.41 (0.08–2.07) | 0.279 | | |
| Perception about chronic diseases | | | | |
| Extremely dangerous | 1 | | | |
| Somewhat or not dangerous | 0.70 (0.08–5.99) | 0.744 | | |
| Able to keep paying OOP | | | | |
| Yes | 1 | | | |
| No | 1.35 (0.26–7.06) | 0.726 | | |
| Aware of the existing health insurance schemes | | | | |
| Yes | 1 | | | |
| No | 2.31 (0.42–12.77) | 0.339 | | |
| Aware of the proposed NHIS | | | | |
| Yes | 1 | | | |
| No | 1.89 (0.36–10.03) | 0.452 | | |

schemes which provide access to preventive and curative health services as well as financing in form of health saving plans and emergency health loans for the poor [19].

The study also found that patients who had been diagnosed with diabetes/hypertension 5–9 years ago were three-fold more likely to enrol in a health insurance scheme compared to those who had been diagnosed with the chronic condition 0–4 years ago. This indicates a positive relationship between length of time since being diagnosed with diabetes/hypertension and acquisition of health insurance. One possible explanation of this finding is that patients with more years since diagnosis may have encountered difficulties paying for health care out of pocket and consequently sought insurance cover. Patients attending the Mbarara Regional Referral Hospital hypertension clinic spend as high as 500,000 shillings per month on medication [2]. In the US, the total estimated cost of diagnosed diabetes in 2012 was estimated at $245 billion, including $176 billion in direct medical costs and $69 billion in reduced productivity [23]. The substantial burden that diabetes/hypertension imposes on patients may thus explain the increased willingness to seek to join insurance schemes as the length of time since diagnosis increases. Similar studies in future should test for this directly by including both the length of time since patients were diagnosed with diabetes/hypertension and the cost of medication as key independent variables.

Other demographic and socio-economic factors including age, gender, marital status, level of education, household size, main source of income, income per month and perception about diabetes/hypertension were not statistically significant.

## Awareness of scheme existence

In this study, utilization of health insurance was found to be significantly associated with awareness about the existence of health insurance schemes. If all factors are held constant, patients who were not aware of the existing schemes were 99% less likely to take up health insurance compared to those who knew about health insurance schemes operating in the study area. This mirrors the findings of studies conducted in Tanzania, Ethiopia and Ghana which reported that awareness of scheme existence was a significant determinant of scheme utilization [24–26]. It is thus crucial that clear messages on health insurance be delivered to patients with diabetes or hypertension possibly through home visits, mass media and awareness campaigns by scheme staff, scheme members and trusted community leaders. In all these awareness-raising initiatives, the focus should not only be on the amount of premium that potential enrolees are expected to pay but should also focus on explaining concepts such as solidarity, optimism, trust and social protection.

## The proposed national health insurance scheme

The vast majority of respondents were supportive of the idea of starting the NHIS in Uganda although concerns were raised about high premiums and poor handling of finances which might ruin trust and hinder individuals from joining the scheme. The challenge of low enrolment due to premiums that are not affordable has previously been reported [27, 28]. In Kenya and Tanzania, previous studies reported that households were not interested in enrolling for health insurance due to corruption [28, 29]. In Uganda, Orem and Zikusooka [30] argued that key systems relating to governance and accountability need to be in place if the NHIS scheme is to be successfully implemented.

The majority of respondents proposed that premiums should be made every month and that each person should, on average, contribute Ug shillings 8,200 (USD 2.3) per month. This amount, however, is considerably high and may be unaffordable for many especially those who have no jobs and those in the informal sector. The amount is even higher than the Ksh

160 (USD 2.0) paid per month in Kenya and the Tsh 5,000–15,000 (USD 2.0–6.0) paid per household per annum in Tanzania [29, 31]. One possible reason why respondents proposed such a high amount to be contributed each month could be that patients with diabetes or hypertension are currently facing a significant burden in paying for medical care so much so that paying about 8,000 shillings per month would represent significant relief. Future studies could investigate this further.

### Study limitations

The major limitations of this study are three-fold. First, the sample of respondents was drawn from a large referral hospital and a private not-for-profit hospital in Mbarara town and may thus be viewed as less suitable to represent the average rural patients in the region. However, these hospitals are referral sites which provide care for many patients with diabetes or hypertension living in other districts in rural southwest Uganda. Second, the predominance of females (n = 235 vs 135 males) might also be seen as a limitation. It might, however, be an indication of gender differences in healthcare-seeking behaviour since men tend to seek care from private for-profit clinics while women more often use government and private not-for-profit health facilities where payment is relatively low [32]. Third, this was a cross-sectional study and as such it was not possible to explore the full range of factors that could influence use of health insurance schemes. Nonetheless, the information generated from the open ended questionnaire could form the basis for a qualitative study to explore other factors this study could have missed.

Despite these limitations, this study, to our knowledge, is the first to demonstrate a relationship between utilization of health insurance and belonging to a microfinance scheme. Our study has important implications for the design of health insurance schemes. First, when designing the forthcoming NHIS the government may use microfinance schemes for priming participants to the behaviour of regular payments. Second, our finding that utilization of health insurance is associated with awareness of scheme existence calls for the government, scheme staff and other stakeholders to design and deliver health insurance messages to the community using various communication channels which may include radio talk shows, home visits and community meetings. Third, it is important that policy makers and health service managers take into consideration the expectations and concerns of people with diabetes and hypertension by ensuring that they get cheaper medical treatment and access quality care.

### Future research

Future studies should seek to gain more in-depth knowledge about microfinance schemes and their potential role as an entry point for health insurance programs for patients in this and other study areas. Another issue for research is whether the cost of accessing medical services influences the willingness of people with chronic diseases to pay relatively high health insurance contributions.

### Conclusions

The findings of this study indicate that utilization of health insurance by patients with diabetes or hypertension is significantly associated with socio-economic factors and awareness about the existence of health insurance schemes. In addition, we found that most of the patients with diabetes or hypertension are willing to enrol in the proposed national health insurance scheme.

## Acknowledgments

We are grateful to the directors of Mbarara Regional Referral Hospital and Divine Mercy Hospital who allowed us to conduct this research within the health facilities. We also wish to thank Otieno Emmanuel, Ghassan K. Abou-Alfa, Chiranjivi Adhikari and an anonymous reviewer for comments which improved our work and our research assistants for their dedication to ensuring high quality work. Our appreciation also goes to the study participants, who willingly gave their time during interviews and shared their unique insights.

## Author Contributions

**Conceptualization:** Peter Kangwagye.

**Data curation:** Peter Kangwagye.

**Formal analysis:** Peter Kangwagye.

**Funding acquisition:** Peter Kangwagye.

**Investigation:** Peter Kangwagye.

**Methodology:** Peter Kangwagye.

**Project administration:** Peter Kangwagye.

**Resources:** Peter Kangwagye.

**Software:** Peter Kangwagye.

**Supervision:** Laban Waswa Bright, Gershom Atukunda.

**Validation:** Peter Kangwagye.

**Visualization:** Peter Kangwagye.

**Writing – original draft:** Peter Kangwagye.

**Writing – review & editing:** Laban Waswa Bright, Gershom Atukunda, Robert Basaza, Francis Bajunirwe.

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
