## [Decision Letter · Decision Letter 0]

26 Dec 2022

PGPH-D-22-00660

“Utilization of health insurance by patients with diabetes or hypertension in urban hospitals in Mbarara, Uganda”

Dear Dr. Kangwagye,

Thank you for submitting your manuscript to PLOS Global Public Health. After careful consideration, we feel that it has merit but does not fully meet PLOS Global Public Health’s publication criteria as it currently stands. Therefore, we invite you to submit a revised version of the manuscript that addresses the points raised during the review process.

We look forward to receiving your revised manuscript.

Kind regards,

Ari Probandari, PhD

Academic Editor

Journal Requirements:

a. Please clarify all sources of funding (financial or material support) for your study. List the grants (with grant number) or organizations (with url) that supported your study, including funding received from your institution. 

b. State the initials, alongside each funding source, of each author to receive each grant.

c. State what role the funders took in the study. If the funders had no role in your study, please state: “The funders had no role in study design, data collection and analysis, decision to publish, or preparation of the manuscript.”

d. If any authors received a salary from any of your funders, please state which authors and which funders.

2. In the online submission form, you indicated that "Data is available via peterkangwagye@gmail.com.". All PLOS journals now require all data underlying the findings described in their manuscript to be freely available to other researchers, either 1. In a public repository, 2. Within the manuscript itself, or 3. Uploaded as supplementary information.

Additional Editor Comments (if provided):

Please respond the reviewers' comments point by point and do necessary revisions.

Reviewers' comments:

Reviewer's Responses to Questions

**Comments to the Author**

1. Does this manuscript meet PLOS Global Public Health’s publication criteria? Is the manuscript technically sound, and do the data support the conclusions? The manuscript must describe methodologically and ethically rigorous research with conclusions that are appropriately drawn based on the data presented.

Reviewer #1: Yes

Reviewer #2: Yes

2. Has the statistical analysis been performed appropriately and rigorously?

Reviewer #1: I don't know

Reviewer #2: Yes

3. Have the authors made all data underlying the findings in their manuscript fully available (please refer to the Data Availability Statement at the start of the manuscript PDF file)?

Reviewer #1: Yes

Reviewer #2: No

4. Is the manuscript presented in an intelligible fashion and written in standard English?

Reviewer #1: Yes

Reviewer #2: Yes

5. Review Comments to the Author

Reviewer #1: Excellent and important effort. Introduction lacks definition and further clarity in regard to microfinance operation. Authors need to take a more thorough look into the process as an outsider. Fixing the definition is key.

Reviewer #2: The authors have reported the utilization of health insurance among a cohort of treatment-seeking patients with diabetes and hypertension in two hospitals in a region of Uganda. The article addresses a pertinent subject of global health importance but needs improvement to improve readability and reproducibility.

Methods

1. Is the questionnaire interviewer-administered or self-administered. Please state

Results

2. The response rate of the respondents ?

3. Define abbreviations at first mention E.g UAP in line 170

4. Line 208-209 is a repetition of the methods

5. When reporting results, the frequencies and then % of the discrete variables should be presented.

6. The tables are not well laid out. They need reworking to allow the rows and columns to align properly

7. Stick with a single appropriate measure of central tendency rather than reporting mean, median and mode.

8. Add legends to abbreviations (NHIS, OOP e.t.c) below the tables

9. report likelihood in ratios rather than percentages

Discussion

The discussion could be improved by exploring the models of micro-health finance schemes in Uganda and how that could enhance service delivery and uptake of policies.

(Aranas LL, Khanam R, Rahman MM, Nghiem S. Combining Microfinance and Health in Reducing Poverty-Driven Healthcare Costs: Evidence From the Philippines. Front Public Health. 2020 Oct 8;8:583455. doi: 10.3389/fpubh.2020.583455. PMID: 33134241; PMCID: PMC7578378.)

Another limitation is that the study method might not be sufficient to explore factors that could influence use of health insurance schemes. The themes generated form the open ended questionnaire could form the basis for a qualitative study to explore other factors this study could have missed.

6. PLOS authors have the option to publish the peer review history of their article (what does this mean?). If published, this will include your full peer review and any attached files.

**Do you want your identity to be public for this peer review?** For information about this choice, including consent withdrawal, please see our Privacy Policy.

Reviewer #1: **Yes: **Ghassan K. Abou-Alfa

Reviewer #2: No

---

## [Decision Letter · Decision Letter 1]

13 Mar 2023

PGPH-D-22-00660R1

“Utilization of health insurance by patients with diabetes or hypertension in urban hospitals in Mbarara, Uganda”

Dear Dr. Kangwagye,

Thank you for submitting your manuscript to PLOS Global Public Health. After careful consideration, we feel that it has merit but does not fully meet PLOS Global Public Health’s publication criteria as it currently stands. Therefore, we invite you to submit a revised version of the manuscript that addresses the points raised during the review process.

The revised manuscript has been evaluated by two reviewers, and their comments are available below.

The reviewers have raised a number of concerns that need attention, and additional information on methodological aspects of the study and analyses are requested.

Could you please revise the manuscript to carefully address the concerns raised?

We look forward to receiving your revised manuscript.

Kind regards,

Vanessa Carels

Staff Editor

Journal Requirements:

Additional Editor Comments (if provided):

Reviewers' comments:

Reviewer's Responses to Questions

**Comments to the Author**

1. If the authors have adequately addressed your comments raised in a previous round of review and you feel that this manuscript is now acceptable for publication, you may indicate that here to bypass the “Comments to the Author” section, enter your conflict of interest statement in the “Confidential to Editor” section, and submit your "Accept" recommendation.

Reviewer #1: All comments have been addressed

Reviewer #3: (No Response)

2. Does this manuscript meet PLOS Global Public Health’s publication criteria? Is the manuscript technically sound, and do the data support the conclusions? The manuscript must describe methodologically and ethically rigorous research with conclusions that are appropriately drawn based on the data presented.

Reviewer #1: Yes

Reviewer #3: Partly

3. Has the statistical analysis been performed appropriately and rigorously?

Reviewer #1: Yes

Reviewer #3: Yes

4. Have the authors made all data underlying the findings in their manuscript fully available (please refer to the Data Availability Statement at the start of the manuscript PDF file)?

Reviewer #1: Yes

Reviewer #3: No

5. Is the manuscript presented in an intelligible fashion and written in standard English?

Reviewer #1: Yes

Reviewer #3: No

6. Review Comments to the Author

Reviewer #1: Thanks for the authors for the thorough review. No further comments

Reviewer #3: Dear Author

Thank you for coming up with the revised manuscript. The abstract, introduction, write up, quantitative inferencing were found in a scientific way. Although you have mentioned in the "data management and analysis" about analysis of qualitative data, this part was not found so well elaborated in the manuscript and needs to be further rectified. The followings are recommended :

1. The tool is mentioned as both for 'quanti' and 'quali' but no detail given about the open-ended/quali part. Kindly describe about the quali-tool under tool subheading. May be the tool be needed to submit under supplemental file.

2. Regarding sample for qualitative part not mentioned elsewhere. Pls add along with qtt samples, number of respondents for qualitative information, you included in the study.

3. For data analysis, also submit the codes, sub-themes, themes in a separate file (supplemental file) and also cite it in your text.

4. The findings does not show up any part of the qualitative aspect, only tables of qtt have been portrayed in results section. So keeping from the verbatims of the respondents should also be included.

5. Data analysis, further needed to triangulate the qtt with qlt so as to infer scientifically.

6. Some recent literatures for the two areas are recommended:

6.1 For SHI and utilization (pls include in discussion, why quality services affect SHI, what are the alternate strategies for SHI to get success, applicabilty provided to in your case of diabetes and hypertension; from the following, but not limited, (only this as an example) to improve the discussion more argumentative.

Paneru DP, Adhikari C, Poudel S, Adhikari LM, Neupane D, Bajracharya J, Jnawali K, Chapain KP, Paudel N, Baidhya N, Rawal A. Adopting social health insurance in Nepal: A mixed study. Frontiers in Public Health. 2022;10.

6.2. relationship of microfinance status and SHI enrollment: a more and rigorous discussion should be added in discussion section, focusing why less people not on microfinance have not enrolled and relationship of not willing to enroll? This part may be explored from the "financial literacy" or any other theoretical background.

For this, it would be wise to add some literatures from recent and theoretical background.

With regards

Reviewer

7. PLOS authors have the option to publish the peer review history of their article (what does this mean?). If published, this will include your full peer review and any attached files.

**Do you want your identity to be public for this peer review?** For information about this choice, including consent withdrawal, please see our Privacy Policy.

Reviewer #1: No

Reviewer #3: **Yes: **Chiranjivi Adhikari

---

## [Editor Report · Decision Letter 2]

19 May 2023

“Utilization of health insurance by patients with diabetes or hypertension in urban hospitals in Mbarara, Uganda”

PGPH-D-22-00660R2

Dear Dr. Kangwagye,

We are pleased to inform you that your manuscript '“Utilization of health insurance by patients with diabetes or hypertension in urban hospitals in Mbarara, Uganda”' has been provisionally accepted for publication in PLOS Global Public Health.

Best regards,

Julia Robinson

Executive Editor